# MRI Assessment of Global and Regional Diaphragmatic Motion in Critically Ill Patients Following Prolonged Ventilator Weaning

**DOI:** 10.3390/medsci7050066

**Published:** 2019-05-22

**Authors:** Khurram Saleem Khan, James Meaney, Ignacio Martin-Loeches, Daniel V. Collins

**Affiliations:** 1Department of Intensive Care medicine, Flinders Medical Centre, Bedford Park, Adelaide, South Australia 5042, Australia; drkhurramsaleem@hotmail.com; 2Department of Radiology, St. James’s Hospital, James’s street, P.O. Box 580 Dublin 8, Republic of Ireland; Jmeaney@meaneys.com; 3Department of Intensive Care Medicine, St. James’s Hospital, James’s street, P.O. Box 580 Dublin 8, Republic of Ireland; jmeaney@meaneys.com; 4Department of Anaesthesia & Intensive Care Medicine, St. James’s Hospital, James’s street, P.O. Box 580 Dublin 8, Republic of Ireland

**Keywords:** ventilator induced diaphragmatic dysfunction (VIDD), magnetic resonance imaging (MRI), weaning, diaphragm displacement and Intensive care unit (ICU)

## Abstract

Introduction: diaphragmatic dysfunction is a common cause of slow weaning in mechanically ventilated patients. Diaphragmatic dysfunction in ventilated patients can be global or regional. The aim of our study was to evaluate the motion of the entire diaphragm in patients who were ventilated for a protracted period in comparison with healthy controls by using Magnetic Resonance Imaging (MRI). Methods: Intensive care patients who had a prolonged ventilator wean and required tracheostomies were enrolled based on extensive exclusion criteria. MRI dynamic sequence and subtraction images were used to measure vertical displacement at five different points on each hemi-diaphragm during normal tidal breathing. Tidal displacement of each point on the right and left hemi-diaphragms of the patients were compared to the precise respective points on the right and left hemi-diaphragms of enrolled controls. Results: Eight intensive care patients and eight controls were enrolled. There were observed significant differences in the displacements of the left hemi-diaphragm between the two groups (median 6.4 mm [Interquartile range (IQR), 4.6–12.5]) vs. 11.6 mm [IQR, 9.5–14.5], p = 0.02). There were also observed significant differences in the displacements at five evaluated study points on the left hemi-diaphragms of the patients when compared to the precise respective points in controls, especially at the dome (median 6.7 mm [IQR, 5.0–11.4] vs. 13.5 mm [IQR 11.5-18], *p* value = 0.005) and the anterior zone of apposition (median 5.0 mm [IQR, 3.3–7.1] vs. 7.8mm [IQR, 7.1–10.5], *p* value = 0.01). The intensive care patients showed lower minimal and maximal values of displacement of right hemi-diaphragms compared to the controls, suggesting that the differences in the displacement of right hemi-diaphragm are possible; however, the differences in the mean values of displacement of right hemi-diaphragm between the intensive care patient group and the control group (median 9.8 mm [IQR (Interquartile range), 5.0–12.3] vs. 10.1 mm [IQR 8.3–18.5], *p* = 0.12) did not reach the level of significance. Conclusion: Although frequently global, diaphragm dysfunction in ventilated patients after prolonged ventilation can also be regional or focal when assessed by MRI dynamic sequence. The vertical displacement of both right and left hemi-diaphragms at various anatomical locations had different values in both controls, and patients. There were significant focal variations in the movement of diaphragm in patients with ventilator-induced diaphragmatic dysfunction.

## 1. Introduction

Diaphragm function is one of the key determinants of the ability to successfully wean patients from mechanical ventilation [1,2]. A large body of evidence from animal models, and more limited data from humans, indicate that mechanical ventilation can cause diaphragmatic muscle fibre injury, a condition called Ventilator-Induced Diaphragmatic Dysfunction (VIDD) [3,4]. Increased oxidative stress and exaggerated proteolysis in the diaphragm have been linked to the development of VIDD in animal models [5,6], but much less is known about the extent to which these phenomena occur in humans undergoing mechanical ventilation in intensive care units. The incidence and prevalence of VIDD and the magnitude of its impact on critically ill patients are not yet fully established. One of the reasons for this problem is that the diaphragm is hard to monitor or evaluate, due to its complex anatomy that continuously changes during breathing. Analysis of the diaphragmatic displacement at locations reveals a gradient that increases from anterior to middle to posterior. Excursion of the lateral aspects is greater than that of the medial aspects [7].

The function of diaphragm is reflected by its position, thickness, and motion. Measuring diaphragm pressure development (i.e., twitch Trans diaphragmatic pressure) is technically demanding, requiring the placement of oesophageal and gastric balloons and special equipment for phrenic nerve stimulation. More recently, studies have shown promising results of the use of ultrasound for monitoring diaphragm function in the ICU [8,9]. Ultrasound visualizes either the dome of the diaphragm or the diaphragm muscle where it abuts the rib cage (the zone of apposition) [9]. Both M-Mode and B-Mode ultrasonography have been used to assess the motion of the dome and the thickness of the diaphragm muscle during inspiration [10]. These measures have been used to determine whether a diaphragm is paralyzed, if atrophy is present, and the suitability of patient to wean from mechanical ventilation. Patients, in the supine position, have been scanned from a low intercostal approach using the liver or spleen as an acoustic window. Numerous studies have assessed the right sided hemi-diaphragm where the liver gives a good acoustic window for access and the left hemi-diaphragm was not consistently visible [9,10].

MRI is a valid tool for the assessment of diaphragm function and movement [11]. The major advantage of MRI lies in its capability of directly acquiring coronal and sagittal images of the entire diaphragm, and this can reveal focal, unilateral, or global diaphragmatic motion abnormalities. The aim of our study was to evaluate the motion of entire diaphragm in patients who were ventilated for a protracted period of time in comparison with healthy controls by using a simple, novel magnetic resonance imaging (MRI) technique.

## 2. Methods

### 2.1. Study Cohort

The study was conducted in a Supra-regional intensive care unit, in a University Teaching Hospital in Dublin, Republic of Ireland from Jan 2010 to Jan 2013. It was approved by St James’s Hospital/Adelaide & Meath incorporating National Children Hospital (SJH/AMNCH) Research Ethics Committee. The reference number is 2009/041/2009/02/Chairman. The study was supported by an educational grant from MSD Ireland.

Eight patients were enrolled on the study following a prolonged period of mechanical ventilation, Median 25 days (IQR 22–27) days [12]. All eight patients had tracheostomies in-situ and were fully liberated from the ventilator before having MRI scans. All patients required inspired fraction of oxygen (FiO2) < 40% by a tracheostomy mask; were off pressure support ventilation or continuous positive airway pressure for at least 24 h; and had no significant electrolyte or metabolic abnormalities, or fever. All patients had muscular power grade using medical research council (MRC) scale of at least 4 for elbow flexion. Each patient had a scan before decannulation of his or her tracheostomy. The patients were ventilated for acute respiratory failure following various illnesses, (Table 2).

The exclusion criteria included morbid obesity, chronic obstructive airway disease, large pleural effusions, lobar collapse, primary neuromuscular disease, ongoing need for intermittent pressure support ventilation, high secretions load requiring frequent suctioning, and patient’s refusal to have an MRI due to claustrophobia. Chest X rays and clinical examination were used to rule out significant lung pathology.

### 2.2. MR Imaging and Analysis of Diaphragm Motion

The study investigator (KSK) and a designated ICU nurse accompanied each patient, during transfer to and from the MRI suite, and into and out of the MRI room along with appropriate emergency equipment and patient monitoring. Written informed consent was obtained from each patient before the study. MRI safety guidelines were observed in addition to filling out an MRI safety screening form. The study investigator (KSK) stayed in the MRI room with the non-control patients to give instructions to take normal breaths and for patient safety reasons. The patients were also visually (using a camera system) and verbally (intercom system) monitored during the scans. None of the patients required any sedation during the scans.

All of the scans were performed during the weekdays between 11:00 to 13:00 h. Unlike most hospital-based scanners that operate at 1.5 Tesla, we employed 3.0 Tesla Philips Achieva MRI system, equipped with 32 RF receiver channels that provide an optimal image uniformity, a high resolution and faster scanning. Image analysis was performed by a Consultant Radiologist (JM) with a special interest in MRI.

Sagittal and coronal dynamic MRI sequences were performed with a balanced FFE (Philips Medical System) approach during tidal breathing. Each image took 0.6 s to acquire, and continuous images were acquired over several respiratory cycles during tidal breathing. Each lung was scanned separately. In order to simplify measurements of diaphragm displacement, the image displaying the diaphragm at its highest position (end expiratory) was used as “mask” from which all other images were subtracted, giving an exact record of diaphragm motion from one slice to another over a temporal timeframe of 600 msec. From the 20 subtraction images generated, the image that represented the greatest amount of displacement of the diaphragm was selected. From that image, the displacement of the hemi-diaphragm was measured in millimetres at 5 different sites (Figure 1). Point 1—4 cm anterior to the point 3; point 2—2 cm anterior to the point 3; point 3—top of the dome of hemi-diaphragm; point 4—2 cm posterior to the point 3; and point 5—4 cm posterior to the point 3. Points 1 and 5 represent the zones of apposition of diaphragm to the chest wall. Each patient therefore had a set of measurements from both their right (R) and left (L) hemi-diaphragms.

### 2.3. Statistical Analysis

We obtained the measurements of vertical displacement in millimetres at 5 specific points on each hemi-diaphragm during tidal breathing. Displacement of each point on the right and left hemi-diaphragms of the patients were compared to the precise respective points on the right and left hemi-diaphragms of the controls. We also measured and compared the mean of 5 values obtained from each hemi-diaphragm. The data is interpreted as median and interquartile range for the displacements of right and left hemi-diaphragms. We log transformed the values and used ‘unpaired t test’ to measure *P* values. A *P* value < 0.05 is considered as significant. We examined the displacements at each point on both hemi-diaphragms to detect a focal abnormality and the mean of 5 measurements on each side to identify global or unilateral motion abnormalities.

## 3. Results

### 3.1. Baseline Characteristics

Eight intensive care patients and eight controls were enrolled. All the controls were male, and one out of eight ICU patients was female. The demographic and clinical characteristics of patients are given in Table 1 and Table 2. The control group has a median BMI of 24.9 (IQR, 24.3–25.5) and a median age of 36 (IQR, 32–43) years. Intensive care patients had a median age of 65 (IQR, 50–78) years, a body mass index (BMI) of 25.5 (IQR, 24.4–26.6), and a median duration of mechanical ventilation of 25 days (IQR, 22–27). The modes of ventilatory assistance included both controlled (Pressure Regulated Volume Controlled ‘PRVC’) and spontaneous modes (Volume Support ‘VS’) of ventilation (Table 1).

### 3.2. MRI Findings

The measurements obtained from the left hemi-diaphragms revealed significant differences in displacements between the two groups (median 6.4 mm [IQR, 4.6–12.2] vs. 11.6 mm [IQR, 9.1–14.5], p = 0.02 (see Table 3). We observed significant differences in the displacements at five evaluated study points on the left hemi-diaphragms of the patients when compared to the precise respective points in controls especially at the dome (L3) (median 6.7 mm [IQR, 5.0–11.4] vs. 13.5 mm [IQR 11.6–18.3], *p* value = 0.005) and anterior zone of apposition (L1) (median 5.0 [IQR, 3.3–7.1] vs. 7.8 [IQR, 7.1–10.5] mm, *p* value = 0.01). The mean values of the right hemi-diaphragm displacement in patients did not differ much from those of controls (median 9.8 mm [IQR, 5.0–12.3] mm vs. 10.1 mm [IQR 8.3–18.5], p = 0.12). The differences in values obtained at different sites from the right hemi-diaphragms of the two groups did not reach the level of statistical significance. However, lower minimal and maximal values of displacement of various sites on right hemi-diaphragms compared to the controls, suggested that the differences in the displacement of right hemi-diaphragm are possible. The difference in displacement of the posterior zone of apposition (R5) of right hemi-diaphragm was close to significance (median 10.3 mm [IQR, 5.5–12.8] vs. 11.9 mm [IQR, 10.4–18.9], *p* value = 0.06). These results indicated that various segments of right and left hemi-diaphragms in both patients and controls have different values of displacement. There were significant focal or regional diaphragm motion abnormalities in our ICU patients group especially on the left side, and displacement of left hemi-diaphragm was significantly more impaired compared to that of right hemi-diaphragm.

## 4. Discussion

This is the first study that has used MRI dynamic sequence and subtraction images to assess diaphragm function in critically ill patients after prolonged mechanical ventilation. MRI has an advantage of examining the entire diaphragm and, although not as widely available as ultrasound, gives more objective evaluation of diaphragm position and function [7,11]. We compared the diaphragm motion of our patients with that of healthy individuals; such a comparison has not been done before in this group of patients. We measured the displacements at five different sites on sagittal views of each hemi-diaphragm, including the areas of diaphragm close to the zones of apposition and the dome of diaphragm. This allowed us to compare motion of all segments of diaphragm in patients with those of healthy individuals.

We selected the patients who had been completely weaned off from the ventilator, required tracheostomies, and were not yet decannulated. This facilitated a safe conduct of MRIs without use of extra sedation and pressure support ventilation that may affect a patient’s respiratory drive, effort, and diaphragm movement. We selected five different sites equally spaced anterior and posterior to the central point on each hemi-diaphragm to measure the focal changes. The measurements of diaphragm displacement during tidal breathing obtained in healthy controls in our study are comparable to those of previous studies using ultrasonography and MRI [13,14,15]. As the diaphragm is composed of peripheral muscular and central tendinous parts, the chosen points represented possibly all regions of the diaphragm viewed on the MRI sagittal and coronal sections. The most anterior and posterior points represent the zones of apposition of diaphragm with the chest wall. Our findings suggest that the left hemi-diaphragm had significantly impaired displacement in patient group when compared to the control group. The observed differences were focal and regional. The right hemi-diaphragm had no significant differences in mean values of vertical displacements between the two groups; however, lower minimal and maximal values in patients compared to the controls indicate a strong possibility of dysfunction of right hemi-diaphragm.

Our findings are subject to a number of limitations. First, as we selected the patients after discontinuation of mechanical ventilation, there is possibility of recovery of some diaphragmatic function during pressure support ventilation and spontaneous breathing [16]. Second, there is an age difference between the two groups; the controls are younger than the patients. There is limited data available in literature to suggest if older age has an impact on diaphragm motion, and there are studies that did not find significant difference in diaphragm thickness on deep breathing and diaphragm motion in quiet breathing amongst different age groups [13,14,15]. Third, MRI cannot be used for routine assessment of diaphragm motion in critically ill patients because of the risks associated with patients transfer and presence of internal and external devices, and monitoring equipment that may not be compatible with MRI. However, it is important to mention that thanks to the development of technology in the last few years, the number of compatible devices in MRI settings and safety has increased. In order to eliminate potential source of error from differences in diaphragmatic position during suspended respiration, we used a dynamic MRI approach that reflected normal tidal breathing. The measurement of diaphragm thickness with a ‘thickening fraction’ during respiratory cycle has been found to demonstrate diaphragmatic dysfunction on ultrasound [17,18,19]. There is currently no evidence to suggest that MRI can be used to measure diaphragmatic thickness. However, this could be of interest in future to have measurements of fractional thickening of all areas of diaphragm simultaneously by using MRI.

## 5. Conclusions

Diaphragmatic dysfunction in ventilated patients is a common cause of weaning failure. Whilst a global assessment might help to identify VIDD, there can be focal or unilateral diaphragmatic dysfunction as assessed by MRI dynamic sequence. Based on our findings, we consider that a prospective larger study should be conducted in order to confirm our findings.

## Figures and Tables

**Figure 1 medsci-07-00066-f001:**
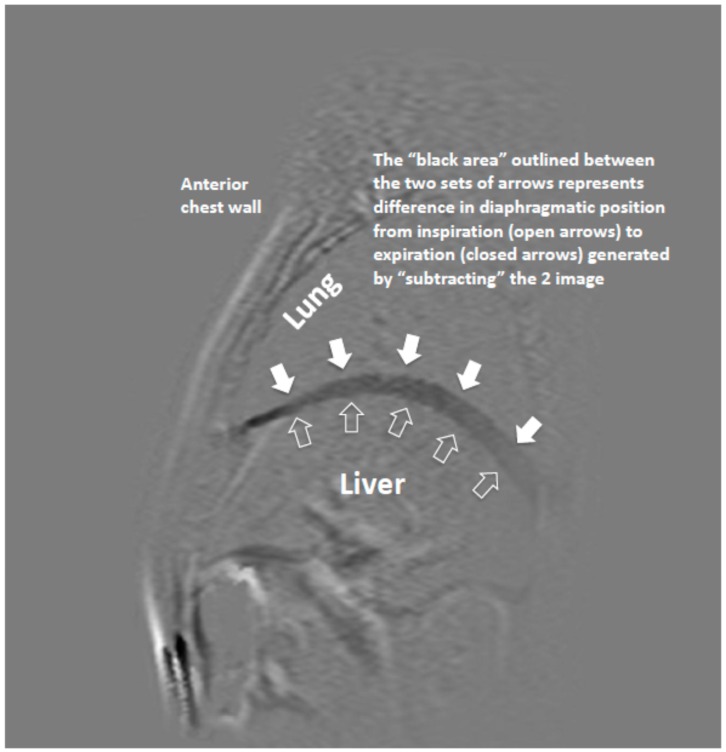
Magnetic resonance (MR) Image of hemi-diaphragm (sagittal view) showing subtraction of maximal and minimal positions of hemi-diaphragm n two sequences.

**Table 1 medsci-07-00066-t001:** Demographics features of Study and Control groups.

Features	Patient Group (Median)	Control Group (Median)
Age	65 (50–78)	36 (32–43)
BMI	25.5 (24.4–26.6)	24.9 (24.3–25.5)
Sex		
Male	7	8
Females	1	
Ventilation (days)		
VS	14 (6–18)	
PRVC	9 (7–20)	
Total	25 (22–27)	
Length of ICU Stay (days)	29 (27–37)	
FiO2	0.32 (0.26–0.4)	
MAP	77 (70–88)	
HR	76 (62–80)	
Temperature	36.6 (36–36.9)	
SpO2	95 (94–96)	

(BMI body mass index, vs. volume support, PRVC pressure regulated volume controlled ventilation, MAP mean arterial pressure, HR heart rate, and FiO2 fractional concentration of inspired oxygen).

**Table 2 medsci-07-00066-t002:** Clinical characteristics of the ICU patient’s group.

Patients	Admission Diagnosis	Comorbidities	Steroid Use
1	Pneumonia	COPD	Yes
2	Pneumonia	CCF	No
3	Pneumonia	NIDDM	No
4	Acute Pancreatitis	NIDDM, AF	No
5	Pneumonia	Hypertension CABG	No
6	Acute Pancreatitis	COPD	Yes
7	Aspiration Pneumonia	None	No
8	Pneumonia	Ischemic heart Disease	No

(COPD–Chronic Obstructive Airways Disease; CCF–Congestive Cardiac Failure; NIDDM–Non-Insulin-Dependent Diabetes Mellitus; CABG–Coronary Artery Bypass Grafting; AF–Atrial Fibrillation).

**Table 3 medsci-07-00066-t003:** Statistical analysis.

Measurement Sites	Patient Median/IQR (mm)	Control Median/IQR (mm)	P Value
**Right hemidiaphragm**			
R1	6.2 (4.1–9.4)	8.0 (5.9–12.2)	0.18
R2	8.9 (4.8–10.7)	9.4 (7–17.9)	0.14
R3	11.1 (5.4–12.8)	11.5 (9–20.9)	0.15
R4	11.6 (5.6–14.5)	11.7 (9.1–21.6)	0.21
R5	10.3 (5.5–12.8)	11.9 (10.4–18.9)	0.06
R (Mean)	9.8 (5.0–12.3)	10.1 (8.3–18.5)	0.12
**Left hemidiaphragm**			
L1	5.0 (3.3–7.1)	7.8 (7.1–10.5)	0.01
L2	5.1 (3.8–9.1)	10.7 (6.4–12.6)	0.02
L3	6.7 (5.0–11.4)	13.5 (11.5–18)	0.005
L4	8.6 (4.3–12.9)	14.3 (11.5–18.8)	0.03
L5	8.1 (3.8–14.7)	12.7 (10.9–17.3)	0.05
L (Mean)	6.4 (4.6–12.2)	11.6 (9.1–14.5)	0.02

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
