# Peer review of "MRI Assessment of Global and Regional Diaphragmatic Motion in Critically Ill Patients Following Prolonged Ventilator Weaning"

_medsci, 2019, doi:10.3390/medsci7050066_

Round 1

Reviewer 1 Report

Presented case control study is referred to MRI assesment of global and diaphragmatic motion in critically ill patetients. Major problem to be considered is the number of patients (and control). As this trial desrcibes 3 years study this group of patients is rather small even if it is a case control study. Another point is that the control group is hardly comparable with the patients group because of age. It would be necessary to fully argument if this kind of control is acceptable in this type of study, also in the case of sex  (both groups are men only with 1 except). In turn appropriate seems the duration of mechanical ventilation time - comparable in all cases.

In presented manuscript there are some technical drawbacks, like: v.167, Table 1a, 8 in no. of males in italic; v.176 Table 1b, Comomorbidities column is not clearly fullfilled, there is some problem with space; v.179 diabetes mellitus not mellitis; MRI findings section - maybe it would be better to present the measurements on the graph?

Reviewer 2 Report

The manuscript “MRI assessment of global and regional diaphragmatic motion in critically ill patients following prolonged ventilator weaning. A case control pilot study.” By Khan und Colleges present a novel approach to asses diaphragm dystrophy resulted from prolonged mechanical ventilation.

It is important to share their findings especially considering the possibility that the current “standard” ultrasound determination of diaphragm thickness is not taking into account potential focal and lateral differences. It also suggest further studies to reveal their origin (anatomic/ physiologic)

 Major concern

#1There is no direct comparison between the IMR and ultrasound estimations in the present study!? The authors used displacement to demonstrate diaphragm weakness. Taking this is the first MRI study it is unlikely that there are known correlation between displacement and thickness and/or strength in this respect. Referring to future studies is somehow not enough to fill this gap :?

#2There was no factor analysis shown to demonstrate that 8 patients and 8 controls are sufficient to demonstrate statistical differences, or otherwise explanation about the group size selection.

#Authors stated that there is no difference in the displacement of the right hemi-diaphragms from healthy individuals and ICU patients based on their t-test p=0.12.  It is however obvious (Table2) that patients showed basically lower minimal and control higher maximal values strongly suggesting that differences in the displacement of right-hemidiaphragm are possible:? I think the authors should be a bit more careful with their conclusions: Next to the long list of limitations, the cohort size may be too small to discern differences in the right hemidiaphragm!

I would suggest, if appropriate for the journal, to limit the findings and statements only to the possibility to apply the MRI technique as a promising tool to investigate potential anatomical differences in the atrophied diaphragm.  

#3It would be interesting to follow and compare the diaphragm displacement changes of patients after complete recovering.  Such data may clarify the role of demographic differences (sex, age, BMI…) etc..

Minor points

Please, introduce all your abbreviations (IQR in the abstract ICU in the introduction; SpO2 (table1)

Please unify your formatting (e.g. the first three paragraphs in the pdf file provided for revision appeared different in letter size; occasional variations in the line distances (L108 M&Ms section)

Line 114 do the authors mean 11:00 to 13:00 :?

Line 115 -116 is a repetition of Line 100-101

Fig.2 appears unnecessary large. It could/should be complemented by visualization of the images showing diaphragm displacement.   

Line 149-151 the explanations at the end of the statistic section belong earlier in the M&Ms description.

Line 158 (n+8) or (n=8) :?

233- is repetition of the introduction section introduction

Reviewer 3 Report

The article presents the assessment of diaphragmatic motion in critically il patients after a long period of mechanical ventilation. Following are the questions and comments that should be addressed:

1-The authors presented the work almost clearly. However, the novelty of the work is not clear. They author claimed the advantages of using MRI for diaphragmatic motion analysis than the ultrasound. But, there are many works done before and showed the same sort of analysis. Such as the works.

"Quantitative analysis of the velocity and synchronicity of diaphragmatic motion: dynamic MRI in different postures." and "Diaphragm Postural Function Analysis Using Magnetic Resonance Imaging".

They also analyzed the motions in different body posture which is another question to the authors.

2- So, if the main contribution of the work is evaluating critically il patients following prolonged ventilator weaning, why the conclusion and core of the discussion are about the merit of MRI than ultrasound.

3- Accordingly, it is better that authors compare and discuss similar works (as pointed above) to the presented article.

4- There are some works done for respiratory monitoring using ultrasound sensors with the following title:

"Ultrasound sensors for diaphragm motion tracking: An application in non-invasive respiratory monitoring"

Could authors compare and evaluate the possibility of this and similar methods in diaphragmatic motion and thickness analysis?

5- Could authors provide a better and detailed picture than Figure 2. Although there is no Figure 1 in this article!!! The picture could be presented with the heat-bar figures. Also, the position of the chest wall is not shown clearly. And, what is the direction of view if a person looks at this image.

6- In the discussion section, lines 280-282, patients are examined after a while. How long after the discontinuation of mechanical ventilation? Could be mentioned in the article.

7- There are many errors in text editing, font size, and punctuations. There is a paragraph right after the Table 1b which seems not relevant to the text of the article. It is an explanation of the abbrevations used in this table. Better to place it as a footnote under the table or caption.

Round 2

Reviewer 3 Report

You can consider it accepted from my side.